# Validation and the associated factors of the Malay version of systemic lupus erythematosus-specific health-related quality of life questionnaires (SLEQoL and LupusQoL)

Nur Aqeelah Ahmad Pouzi[1], Syahrul Sazliyana Shaharir[1]*, Azmi Mohd Tamil[2], Ruslinda Mustafar[1‡], Suhaida Ahmad Maulana[3‡], Eashwary Mageswaren[3‡], Wan Syamimee Wan Ghazali[4]

1 Department of Internal Medicine, Faculty of Medicine, Universiti Kebangsaan Malaysia, Jalan Yaacob Latiff, Cheras, Kuala Lumpur, Malaysia, 2 Department of Public Health, Faculty of Medicine, Universiti Kebangsaan Malaysia, Jalan Yaacob Latiff, Cheras, Kuala Lumpur, Malaysia, 3 Medical Department, Hospital Tengku Ampuan Rahimah, Klang, Selangor, Malaysia, 4 Department of Internal Medicine, School of Medical Sciences, Health Campus, Universiti Sains Malaysia, Kubang Kerian, Kelantan, Malaysia

☯ These authors contributed equally to this work.
‡ RM, SAM, and EM also contributed equally to this work.
* sazliyana@ukm.edu.my

## Abstract

### Objectives

To assess the reliability and validity of two disease-specific questionnaires that assess the quality of life (QoL) among patients with Systemic Lupus Erythematosus (SLE); SLEQoL and LupusQoL in Malay language. This study also identified the factors affecting each domain of the questionnaires.

### Methods

This cross-sectional study was conducted from June 2021 until April 2022, and SLE patients were recruited to complete the SLEQoL, LupusQoL and Short Form Health Survey (SF-36) in Malay language. Disease activity were recorded using the modified SLE Disease Activity Index (M- SLEDAI) and British Isles Lupus Assessment Group 2004 (BILAG-2004) index. Presence of organ damage was determined using the SLICC Damage index. Cronbach's alpha was calculated to determine internal consistency while exploratory factor analysis was done to determine the construct validity. Concurrent validity was evaluated using correlation with SF-36. Multiple linear regression analysis was deployed to determine the factors affecting each domain of SLEQoL and LupusQoL.

### Results

A total of 125 subjects were recruited. The Cronbach's α value for the Malay-SLEQoL (M-SLEQoL) and Malay-LupusQOL (M-LupusQoL) was 0.890 and 0.944 respectively. Exploratory factor analysis found formation of similar number of components with the original

**Data Availability Statement:** All relevant data are within the paper and its Supporting information files.

**Funding:** The author(s) received no specific funding for this work.

**Competing interests:** The authors have declared that no competing interests exist.

version of questionnaires and all items have good factor loading of >0.4. Both instruments also had good concurrent validity with SF-36. M-SLEQoL had good correlations with BILAG-2004 and M-SLEDAI scores. Musculoskeletal (MSK) involvement was independently associated with lower M-SLEQoL in physical function, activity and symptom domains. Meanwhile, MSK and NPSLE were associated with fatigue in M-LupusQoL.

## Conclusion

Both M-SLEQoL and M-LupusQoL are reliable and valid as disease -specific QoL instruments for Malaysian patients. The M-Lupus QoL has better discriminative validity compared to the M-SLEQoL. SLE patients with MSK involvement are at risk of poor QoL in multiple domains including physical function, activity, symptoms and fatigue.

## Introduction

Systemic Lupus Erythematosus (SLE) is a complex multisystem autoimmune disease with a wide spectrum of the clinical phenotype. The course and outcomes of SLE are influenced by several factors including genetic, environmental and sociodemographic background [1]. Despite the improvement of survival due to the advances in the treatment of SLE, their quality of life (QoL) is negatively affected due to the substantial morbidity from the disease and the treatment itself. It has been demonstrated that QoL is reduced in patients with SLE compared with patients Sjögren's syndrome and rheumatoid arthritis [2], and with the general population [3]. Recognizing the importance of measuring QoL in SLE, the Outcome Measures in Rheumatology (OMERACT) group [4] and various regulatory agencies including European Medicines Agencies (EMA) and the United State Food and Drugs Administration (U.S. FDA) recommend that all drugs trial should include patient-reported outcome (PRO) as one of the therapeutic outcomes [5, 6].

However, the PRO tools to assess QoL in SLE are mostly designed in English and hence mainly being validated among English-speaking population [7]. Currently, in Malaysia, there is lack of a valid and reliable PRO instrument for our Malaysian SLE patients. So far, only the generic Medical Outcomes Study Short-Form 36 (SF36 Malay version) questionnaire has been validated in Malaysia [8]. Employing the generic tool to assess QOL in SLE may not adequately capture symptoms or issues that are specific to the disease [9]. Therefore, there is an urgent need to validate the PRO tools among our Malaysian SLE population to adapt with the local linguistic and cultural differences. Validated PRO tool in different languages enable more participation of patients from all over the world in randomized controlled trials (RCTs) to prevent study bias by lack of recruitment of patients from the non-English speaking countries in clinical trials [10].

The most-widely used SLE disease-specific PRO tool is LupusQoL which was developed in the UK [11]. LupusQoL has been widely validated and adapted to other cohorts and has been translated into 77 languages for use in 51 countries. Itis more being widely used in randomized–controlled clinical trials (RCTs) [9]. SLEQoL was initially developed and validated in English language among Singaporean Chinese patients. This tool incorporates questions suitable for oriental cultures [12], and since Malaysia is a neighbour country of Singapore, the content may be more culturally adapted with our population. It consists of 40-items and it demonstrates better responsiveness to change than the SF-36. The Chinese version of SLEQoL

has also been validated in Singapore [13] and in addition, it has been translated and validated in Thailand [14] and Egypt [15] using their native languages.

There are various factors that can affect the QoL in SLE patients [16] and literature on QoL among Malaysian SLE population is still sparse. This is likely due to no disease-specific tool that is available in Malay language. Hence, there is an urgent need to validate the QoL assessments tool in Malay language, to encourage clinician to use these tools to assess the QoL aspect of patients. The validation may also enable more Malay patients who are not proficient in English language to participate in the international RCTs. In addition, this will also encourage the clinician to use these validated instruments and thus will further enhance the holistic approach of the management of the SLE patients in Malaysia.

In this study, we aim to determine the validity and reliability of two disease-specific patient reported outcome (PRO) instruments; SLE Quality of Life (SLEQoL) and Lupus Quality of Life (LupusQoL) in Malay language among SLE patients in two major Rheumatology centres. In addition, this study also evaluates the correlations and associations between the PRO instruments with SF-36, socio-demographic characteristics, SLE disease activity, presence of co-morbidities and presence of organ damage.

## Materials and methods

### Patients

This study was a cross-sectional study conducted in Hospital Canselor Tuanku Muhriz (HCTM) Universiti Kebangsaan Malaysia and Hospital Tengku Ampuan Rahimah (HTAR) Klang from June 2021 until April 2022. Adult SLE patients ($\geq$ 18 years old) who fulfilled the American College Rheumatology (ACR) 1997 classification criteria [17] and/or Systemic Lupus Erythematosus Collaborating Clinics (SLICC) 2012 classification criteria [18] were recruited. Patients that were included have to be proficient in spoken and written Malay language. Patients were recruited from the outpatient clinics and convenience sampling method was used to recruit patients.

Patients with documented severe cognitive impairment (dementia, Alzheimer's disease), psychiatric illness, and permanent physical disability not related to SLE or had an overlap SLE with other connective tissue diseases were excluded. Written informed consent was obtained from all patients and this study obtained approval from the UKM Ethics Committee (JEP-2021-332) and National Medical Research Register (NMRR DI-21-02263-DCN).

**Data collection.** While SLE disease characteristics were majorly extracted from patients' medical records i.e. age of onset, duration of disease, organ or system involvement, and co-morbidities (hypertension, diabetes mellitus, ischemic heart disease), some sociodemographic data was obtained (e.g. age, gender, ethnicity, level of education, and type of occupation) via self-reporting google form or during clinical interviews. Patients self-completed the Malay version of SLEQoL (M-SLEQoL), LupusQoL (M-LupusQoL) and SF-36 (M-SF36) using hardcopies or google form.

**Sample size calculation.** All of the questionnaires (SF-36, M-SLEQoL and M-LupusQoL) that were used in this study have more than 5 loadings per item and based on the highest loading per item in M-SLEQoL (40 items), the calculated sample size was 200 patients [19].

**SLE disease activity and damage.** SLE disease activity assessments were performed at study visit in all patients using the modified-Systemic Lupus Erythematosus Disease Activity Index (M-SLEDAI) and British Isles Lupus Assessment Group Index (BILAG 2004) score [20]. Modified-SLEDAI which omits the serology items (complement levels, anti-dsDNA antibodies) is a validated and reliable instrument for disease activity assessments [21]. Total BILAG 2004 index scores were calculated based on category i.e. A = 12, B = 8, C = 1 and D/E = 0 [22].

Presence of irreversible organ damage was determined using the Systemic Lupus International Collaborating Clinics/American College of Rheumatology (SLICC/ACR) Damage Index (SDI) [23]. Patients were divided into 2 groups: those with damage (SDI score ≥1) and those without damage (SDI score = 0). Both disease activity and organ damage assessments were verified by a Rheumatologist who was blinded to the cases.

## Validation of the Malay version of SLEQoL and LupusQoL (M-SLEQoL and M-LupusQoL)

The validation of the Malay version of SLEQoL and LupusQoL (M-SLEQoL and M-Lupus-QoL) and consisted of 5 stages:

1. Content, translation and face validity

2. Internal consistency reliability by determining the Cronbach's-α [24].

3. Construct validity using exploratory factor analysis [25]

4. Concurrent validity with the generic tool Short Form Health Survey in Malay language (M-SF36)

5. Discriminant validity with disease activity and damage

**Content, translation and face validity of the M-SLEQoL and M-LupusQoL.**   Both of the original version of SLEQoL and LupusQoL questionnaires are in English language and they have been translated to Malay language by the original authors according to the industrial standard. However, the Malay version of SLEQoL (M-SLEQoL) and LupusQoL (M-Lupus-QoL) have never been validated.

Content validity was performed by the expert panels of the researchers which consists of 2 rheumatologists, and an academician public health specialist experienced in the validation of instruments. They have reviewed the content and decided that all the items were relevant and important. The Item-content validity index (I-CVI) and scale content validity index of both questionnaires were 1 (S1 File). The expert panels have also reviewed the translated version of the questionnaires and found that they were similar as the original version.

Face validity was done through cognitive debriefing of the questionnaires in 10 healthy subjects and 10 SLE patients to determine items clarity and comprehensibility. They were asked for their views regarding the phrasing and content of the instruments. All subjects understood the translated version and did not encounter problems in answering the questionnaire. However, we did not record the subjects' rates on the items in the questionnaires and hence the Item-face validity index (I-FVI) and (Scale-face validity index) S-FVI were not calculated.

We have obtained permission from the original authors for the validation process. Table 1 compares the domains, response format and score interpretation of SLEQol and LupusQoL questionnaires.

**Table 1. Comparisons between Lupus QoL and SLE QoL.**

| Measure | Recall period | Domains | Response format | Range Of Scores | Score interpretation |
|---|---|---|---|---|---|
| Lupus QoL [11] | 4 weeks | 34 items grouped into 8 domains: Physical health, emotional health, body image, pain, planning, fatigue, intimate relationships, and burden to others | 5 -point Likert scale | 0–100 | Higher score indicates better quality of life |
| SLE QoL [12] | 1 week | 40 items grouped into 6 domains: physical functioning, activities, symptoms, treatment, mood, and self-image | 7-point Likert scale | 40–280 | Higher score indicates worse quality of life |

Subsequently the Malay versions of both questionnaires (M-SLEQoL and M-LupusQoL) were used in SLE patients for further testing of their reliability and validity. The subjects also completed the Malay version of the Short Form Health Survey (M-SF36) which has been validated and extensively used in patients with chronic diseases including SLE in Malaysia [8, 26–29].

## Statistical analysis

Data were analysed using the IBMM SPSS software (version 26). Kolmogorov Smirnov test was performed to assess normality of the data and the results showed that all variables had p<0.001, indicating that data was not normally distributed. Therefore, non- parametric statistical analysis methods were used.

Cronbach's-α was performed to test the internal consistency of the questionnaires and was considered good if > 0.7 [24]. To determine the construct validity, exploratory factor analysis (EFA) was conducted to extract the factor structure of the questionnaires [25]. Prior to EFA analysis, Kaiser-Meyer-Olkin (KMO) and Bartlett's tests were done to ensure sampling adequacy and suitability. The sampling is adequate if the value of KMO test is more than 0.5 while for Bartlett's test, the significant value of less than 0.05 is acceptable for further analysis. The six factors in M-SLEQoL and 8 factors in M-LupusQoL were computed based on the factor extracted by the original questionnaires. The factors were rotated with Varimax rotations so that each variables loads only on one factor/component. This was an orthogonal method that ensures that the factors are uncorrelated. Variables with factor loadings of 0.4 and above were acceptable and were considered stable [30].

Concurrent validity was determined by performing bivariate Spearman's rho test with SF36 while discriminant validity with disease activity scores (M-SLEDAI and BILAG-2004) and damage index (SDI). Spearman's correlation $r_s$>0.7 is regarded as showing very strong positive correlation; $r_s$ from 0.4 to 0.69 suggests strong positive correlation, $r_s$ from 0.30 to 0.39 suggests moderate positive correlation, $r_s$ from 0.20 to 0.29 suggests weak positive correlation, $r_s$ from 0.19 to −0.19 suggests no or negligible correlation, $r_s$ from −0.20 to −0.29 suggests weak negative correlation, $r_s$ from −0.30 to −0.39 suggests moderate negative correlation, $r_s$ from −0.40 to −0.69 suggests strong negative correlation and −0.70 or lower suggests very strong negative correlation.

To determine factors associated with QoL, Mann Whitney test and Kruskal Wallis test were conducted for 2 categorical and ≥ 3 categorical variables respectively and Spearman's test for continuous data. Subsequently, multivariate linear regression analysis which included all variables with p value of <0.10 was performed to determine the independent predictors of each QoL domain. In all analyses, a P value (two-tailed) of less than 0.05 was considered statistically significant.

## Results

### Socio-demographics and disease characteristics

A total of 125 patients were recruited and this study cohort was comprised of predominantly females (92%, n = 115) and Malay ethnicity (83.2%, n = 104). Majority of the patients were female n = 115(92%) while only n = 10(8%) were male patients. Majority were in the low socioeconomic category (household income less than Malaysian Ringgit, MYR 4,849/month or USD1,018/months). Table 2 illustrates the baseline socio-demographics and disease characteristic of the SLE patients.

**Table 2. Socio-demographics and disease characteristics of the SLE patients.**

| Characteristics | Frequency, n (%) |
|---|---|
| Age, years | 36.4 ± 11.7 |
| Disease duration, years | 7.6 ± 6.8 |
| Gender | |
| Female | 115 (92%) |
| Male | 10 (8%) |
| Race | |
| Malay | 104 (83.2%) |
| Chinese | 13 (10.4%) |
| Indian | 6 (4.8%) |
| Others | 2 (1.6%) |
| Religion | |
| Muslim | 106 (84.8%) |
| Buddhist | 10 (8%) |
| Hindu | 4 (3.2%) |
| Christian | 4 (3.2%) |
| Education level | |
| Higher education | 70 (56%) |
| Secondary education | 55 (44%) |
| Marital status | |
| Married | 77 (61.6%) |
| Single | 38 (30.4%) |
| Divorced | 8 (6.4%) |
| Widow | 2 (1.6%) |
| Employment | |
| Unemployed | 45 (36%) |
| Employed | 80 (64%) |
| Household Income | |
| Less than MYR 4,849 (low income) | 100 (80%) |
| More than MYR 4,850 | 25 (20.0%) |
| System involvement | |
| Musculoskeletal | 88 (70.4%) |
| Haematological | 47 (37.6%) |
| Mucocutaneous | 62 (29.6%) |
| Renal | 37 (29.6%) |
| Neuropsychiatric lupus (NPSLE) | 16 (12.8%) |
| Comorbidities | |
| Hypertension | 28 (22.4%) |
| Dyslipidemia | 15 (12%) |
| Diabetes mellitus | 7 (5.6%) |
| Heart disease | 6 (4.8%) |
| M-SLEDAI score | 0 (0–19) |
| BILAG-2004 score | 0 (0–60) |
| SDI damage index | |
| Presence of damage (SDI≥1) | 23(18.4%) |

M-SLEDAI = Modified SLE Disease Activity Index, BILAG-2004 = British Isles Lupus Assessment Group 2004 (BILAG-2004), SDI = Systemic Lupus International Collaborating Clinics/ American College of Rheumatology damage index, MYR = Malaysian Ringgit

## Content, translational and face validation of the SLEQoL and LupusQoL in Malay

After review of the content and translated version of SLEQoL and LupusQoL, no changes were made with the translated version (M-SLEQoL and M-LupusQoL) provided by the original authors (S1 Appendix-SLEQoL Malay version and S2 Appendix-LupusQoL Malay version).

During the cognitive debriefing of the questionnaires, all subjects concluded that the M-SLEQoL and M-LupusQoL were understandable and no changes were made. The subjects took about 5–10 minutes to complete the questionnaire. The subjects also completed the English version and no difference was found between the Malay and original version of the questionnaires.

Analysis of the 125 SLE patients who completed the M-SLEQoL and M-LupusQoL revealed that the data were not distributed normally by Kolgomorov-Smirnov test with p < 0.05. Table 3 shows the 8 domains of Lupus QoL and 6 domains of SLE QoL and their median values.

## Internal consistency

The Cronbach's α value for M-SLEQOL Malay was 0.890 for all 6 domains which indicates good internal consistency. The Cronbach's α for domain physical function (PFSQ) was 0.938, activities (ACTVSQ) 0.912, symptoms (SYMPSQ) 0.897, treatment (TRSQ) 0.750 and mood (MOODSQ) 0.926 and self-imaged IMAGESQ (0.907).

The Cronbach's α value for all 8 domains in M-LupusQOL was 0.944, which indicates good internal consistency. Cronbach's α for physical function (PFLQ) was 0.957, pain (PAINLQ) 0.916, planning (PLANLQ) 0.968, intimate relationship (INTRELLQ) 0.993, burden to others (BURDENLQ) 0.942, emotional health (EMOLQ) 0.973, body image (IMAGELQ) 0.909 and fatigue (FATIGUELQ) 0.884.

S1 and S2 Tables illustrates the Cronbach's alpha for the total domain and its sub-items of M-SLEQoL and M-LupusQoL respectively.

## Construct validity using Exploratory Factor Analysis (EFA)

The Kaiser-Mayer-Olkin (KMO) and Bartlett's test of sphericity (p) for M-SLEQoL was 0.894, p < .001 while M-LupusQoL was 0.946, p<0.001, which indicates sampling was adequate and suitable for EFA. The factor loadings of all the items in M-SLEQoL and M-LupusQoL were acceptable (>0.4).

For M-SLEQoL, the results from the EFA showed 6 components with the first 8 factors Eigen values were greater than 1.0 and the value of the total variance explained by these eight factors was 70.487% that determine number of factors to become domains. Using factor

**Table 3. The domains of M-LupusQoL and M-SLEQoL and their median values.**

| LupusQoL domains | Median (IQR, min-max) | SLEQoL Domains | Median (IQR, min-max) |
|---|---|---|---|
| Physical function | 75 (IQR 50, 0–100) | Physical function | 9 (IQR 8.5, 6–39) |
| Pain | 75 (IQR 58.3, 0–100) | Symptoms | 16 (IQR 12.5, 8–55) |
| Image | 70 (IQR 50, 0–100) | Self-image | 21 (IQR 18, 9–62) |
| Intimate relationship | 75 (IQR 100, 0–100) | Treatment | 7 (IQR 18, 4–20) |
| Planning | 83.3 (IQR 70.8,0–100) | Activity | 19 (IQR 19, 9–59) |
| Emotional | 70.8 (IQR 64.5.0–100) | Mood | 9 (IQR 10.5, 4–28) |
| Fatigue | 62.5 (IQR 50, 0–100) | | |
| Burden | 66.7 (IQR 66.7, 0–100) | | |

analysis, M-SLEQoL 6 domains physical function (PFSQ), activity (ACTVSQ), symptom (SYMPSQ), treatment (TRSQ), mood (MOODSQ), self image (IMAGESQ) were tested.

6 factors were extracted which represent 6 domains in M-SLEQoL with good factor loading for each item. S3 Table illustrates the Eigen values for M-SLEQoL.

For M-LupusQoL, results from the EFA showed 8 components with the first 4 factors Eigen values greater than 1.0 while the value of the total variance explained by these eight factors was 87.705% that determine the number of factors to become domains. Using factor analysis, M-LupusQOL comprised of 8 domains physical function (PF), pain (PAINLQ), planning (PLANLQ), intimate relationship (INTRELLQ), burden (BURDENLQ), emotion(EMOLQ), image (IMAGELQ) and fatigue (FATIGUELQ) were tested. 8 factors were extracted which represent 8 domains in M-LupusQOL with good factor loading for each item. S4 Table illustrates the Eigen values for M-LupusQoL.

In the rotated component matrix for M-SLEQoL questionnaire, items for domain physical function (PF1 to PF6) were loaded into the same group. Mood domain (MOOD1 until MOOD4) and self-image domain (IMAGE1 until IMAGE8) were loaded into the same group, except item IMAGE9. Items for domain activity (ACTV1 to ACTV 8) were loaded in the same component, except for item ACTV9. Items for treatment (TREATMENT1 to TREATMENT4) were grouped in the same component while items for domain symptom SYMP1 and SYMP4-8 were found in the same component, except for SYMP2 and SYMP3. Component 6 comprised of 2 items ie TR1 and IMAGE9. S5 Table illustrates the factor analysis of 40 items in M-SLEQoL.

In the rotated component matrix of M-LupusQOL, all items in the physical function domain (PF1-PF8), pain (PAIN1-3) and planning (PLAN1-3) were loaded into the same component. All items in the emotion domain (EMO1 to EMO6) were loaded in the same component 2, and all items in the burden domain (BURDEN1 to BURDEN3) were in the same component 3. All items in image domain were loaded in the same component number 4 except for item IMAGE4. All items in fatigue domain were grouped into the same component except for item FATIGUE3. Inter-relationship items were loaded in the same component 6. S6 Table illustrates the factor analysis of 34 items in M-LupusQoL.

## Concurrent validity of M-LupusQoL and M-SLEQoL with M-SF36

There were moderate positive correlations between the M-LupusQoL domains physical function, fatigue and pain with the comparative domains of M-SF36 ($r_s$ ranging from 0.3 to 0.37). A stronger correlation was found in the emotion domain, as illustrated in Table 4.

There were strong negative correlations ($r_s$ ranging from −0.50 to −0.61) between the four equal domains in M-SLEQoL and M-SF36, as illustrated in Table 4.

## Discriminant validity and correlation with disease activity and damage

There were weak correlations between physical function, activity, treatment, image, mood domain scores of M-SLEQoL with BILAG-2004 and M-SLEDAI score ($r_s < 0.3$). There was no correlation with damage or SDI score, as illustrated in Table 5. There was no correlation between M-LupusQoL with disease activity and damage index.

## The associated factors of M-SLEQoL

In univariate analyses, there were inverse correlations between age and treatment, image, mood domains and total M-SLEQoL, suggesting that younger patients had worse QoL. Disease duration was negatively correlated with physical function, activity, image, and mood domains, which indicate lower QoL in these domains among patients with longer disease duration.

**Table 4. Correlation between the comparable domain in M-SLEQoL and M-LupusQoL with M-SF36.**

| SLEQoL domain | Median (IQR) | M-SF36 domain | Median (IQR, min-max) | r$_s$ (p value) |
|---|---|---|---|---|
| Physical function | 9 (IQR 8.5 | Physical function | 70 (IQR 43, 0–100) | -0.538, <0.001 |
| Self-image | 21 (IQR 18) | Social functioning | 75 (IQR 50, 0–100) | -0.610, <0.001 |
| Mood | 9 (IQR 10.5 | Emotional health | 70 (IQR 43, 0–100) | -0.563, <0.001 |
| Symptoms | 16 (IQR 12.5 | Bodily pain | 68 (IQR 44, 0–100) | -0.600, <0.001 |
| LupusQoL domain | Median (IQR) | SF 36 domain | Median (IQR, min-max) | r$_s$ (p value) |
| Physical function | 75 (IQR 50) | Physical function | 70 (IQR 43, 0–100) | 0.307, <0.001 |
| Fatigue | 62.5 (IQR 50 | Vitality | 55 (IQR 23,1–100) | 0.366, <0.001 |
| Emotion | 70.8 (IQR 64.5 | Emotional health | 70 (IQR 43, 0–100) | 0.493, <0.001 |
| Pain | 75 (IQR 58.3 | Bodily pain | 68 (IQR 44, 0–100 | 0.377,<0.001 |

r$_s$ = Correlation coefficient

Patients with hypertension and musculoskeletal involvement had worse physical function domain (p value 0.05 and 0.07, respectively). In contrast, Muslim patients had better QoL in physical function domain (p<0.05). Musculoskeletal (MSK) involvement was associated with worse QoL in activity domain while Malay and Muslim patients had better QoL in this domain (all P<0.05). Worse symptoms domain was associated with both mucocutaneous and MSK involvement while worse QoL in the image domain was associated with lower income category, being married and haematological involvement (all p<0.05).

In the multivariate linear regression analysis which included all variables with p value of <0.10 in the univariate analysis revealed that MSK involvement was independently associated with lower QoL in physical function, activity and symptom domains. Meanwhile, BILAG-2004 scores were correlated with worse QoL in physical and image domains. Table 6 illustrates the multivariate analysis of the independent factors associated with the different domains in M-SLEQoL.

## The associated factors of M-LupusQoL

Patients with musculoskeletal (MSK) involvement had worse pain and planning QoL scores [66.67(IQR66.67) vs 83.3(IQR45.83), p = 0.017] and [75(IQR75) vs 100(IQR41.67), p = 0.01], respectively. Musculoskeletal involvement remained as a significant factor in the multivariate linear regression analysis which included other variables that were significant at p<0.10 in the initial univariate analysis.

Patients with MSK, muco-cutaneous, and neuropsychiatric lupus (NPSLE) tend to have lower scores in the fatigue domain, with p<0.1. In the multivariate linear regression analysis, MSK and NPSLE remained to be significantly correlated with low fatigue domain scores.

Patients with renal involvement had better intimate relationship QoL score in the univariate analysis [100 (IQR 62.5) vs 31.25(IQR100), p = 0.002]. However, it was no longer significant in the multivariate linear regression analysis. No associations or correlations were found with other socio-demographic, disease characteristics and patients' co-morbidities.

## Discussion

The assessment of QoL has become an essential component apart from the disease remission and prevention of organ damage in SLE. Therefore, patient reported outcome (PRO) instruments are designed to assess their QoL. QoL affects several aspects of the patients' lives including their work [31], daily-living activities [28] and their coping mechanisms [32]. By exploring

**Table 5. Correlation between M-SLEQoL and M-LupusQoL with disease activity and damage.**

| Domain | $r_s$/ p value | BILAG-2004 | M-SLEDAI | SDI score |
|---|---|---|---|---|
| M-SLEQoL | | | | |
| Physical function | $r_s$ | 0.24 | 0.24 | 0.10 |
| | p | 0.007* | 0.007* | 0.28 |
| Activity | $r_s$ | 0.29 | 0.29 | 0.05 |
| | p | 0.001* | 0.001* | 0.61 |
| Symptom | $r_s$ | 0.15 | 0.13 | -0.06 |
| | p | 0.11 | 0.14 | 0.87 |
| Treatment | $r_s$ | 0.18 | 0.17 | 0.02 |
| | p | 0.04* | 0.06 | 0.87 |
| Self-Image | $r_s$ | 0.18 | 0.19 | 0.02 |
| | p | 0.04* | 0.03* | 0.83 |
| Mood | $r_s$ | 0.22 | 0.23 | 0.02 |
| | p | 0.02* | 0.01* | 0.84 |
| M-LupusQoL | | | | |
| Physical function | $r_s$ | -0.09 | -0.02 | -0.06 |
| | p | 0.35 | 0.23 | 0.52 |
| Pain | $r_s$ | -0.06 | -0.06 | -0.08 |
| | p | 0.54 | 0.48 | 0.39 |
| Planning | $r_s$ | -0.07 | -0.07 | -0.10 |
| | p | 0.46 | 0.43 | 0.27 |
| Inter-relationship | $r_s$ | 0.07 | 0.08 | -0.07 |
| | p | 0.44 | 0.36 | 0.46 |
| Burden | $r_s$ | -0.16 | -0.16 | -0.09 |
| | p | 0.08 | 0.09 | 0.31 |
| Emotion | $r_s$ | -0.09 | -0.10 | -0.06 |
| | p | 0.30 | 0.25 | 0.53 |
| Image | $r_s$ | -0.03 | -0.05 | 0.001 |
| | p | 0.72 | 0.58 | 0.99 |
| Fatigue | $r_s$ | 0.02 | 0.006 | -0.02 |
| | p | 0.85 | 0.95 | 0.79 |

*significant p value at <0.05.

$r_s$ = Correlation coefficient

BILAG-2004 = British Isles Lupus Assessment Group index 2004, M-SLEDAI = Modified-SLE Disease Activity index, SDI = SLICC damage index

the multi-dimensional aspects of the patients, this will facilitate better communication with the multidisciplinary team to optimize their health status [33].

Lupus specific measures are superior to generic scales in the assessing the QoL [34].

Our study was the first in Malaysia to validate lupus-specific QoL instruments and to assess the impact of demographic and disease-related characteristics on HRQoL using the validated translated Malay version of SLEQoL and LupusQoL (M-SLEQoL and M-LupusQoL). In our study, both M-SLEQoL and M-LupusQoL were demonstrated to have a good internal consistency reliability.

Both instruments also have concurrent validity with the comparable individual domains in the generic QoL SF-36. M-SLEQoL demonstrates stronger correlations with SF36 and this finding is in contrast with the initial development and validation of SLEQoL, which showed poor concurrent validity with the SF-36 [12]. On the other hand, the initial development and

**Table 6. Multivariate linear regression analysis of the factors associated with different domains in M-SLEQoL.**

| Domain | Unstandardized Coefficients | | Standardized Coefficients | t | 95.0% Confidence Interval for B | | P value |
|---|---|---|---|---|---|---|---|
| | B | Std error | β | | Lower bound | Upper bound | |
| **Physical function** | | | | | | | |
| BILAG-2004 | 0.48 | 0.21 | 0.49 | 2.33 | 0.07 | 0.89 | 0.02* |
| M-SLEDAI | -0.42 | 0.49 | -0.18 | -0.85 | -1.41 | 0.56 | 0.39 |
| Disease duration | -0.09 | 0.10 | -0.08 | -0.97 | -0.29 | 0.10 | 0.34 |
| Muslim | -2.76 | 1.92 | -0.12 | -1.44 | -6.56 | 1.05 | 0.15 |
| Hypertension | -2.70 | 1.65 | -0.14 | -1.64 | -5.97 | 0.57 | 0.10 |
| Musculoskeletal | 3.22 | 1.46 | 0.19 | 2.21 | 0.34 | 6.09 | 0.03* |
| **Activity** | | | | | | | |
| BILAG-2004 | 0.58 | 0.32 | 0.36 | 1.79 | -0.06 | 1.22 | 0.08 |
| M-SLEDAI | -0.07 | 0.78 | -0.02 | -0.09 | -1.61 | 1.47 | 0.93 |
| Age | -0.05 | 0.10 | -0.05 | -0.52 | -0.24 | 0.14 | 0.61 |
| Duration | -0.19 | 0.17 | -0.10 | -1.13 | -0.52 | 0.14 | 0.26 |
| Musculoskeletal | 7.14 | 2.32 | 0.26 | 3.08 | 2.60 | 11.73 | 0.003* |
| Muslim | -10.36 | 8.59 | -0.29 | -1.21 | -27.37 | 6.66 | 0.23 |
| Malay | 4.014 | 8.228 | 0.12 | 0.49 | -12.28 | 20.31 | 0.63 |
| **Symptom** | | | | | | | |
| Mucocutaneous | 2.17 | 1.68 | 0.11 | 1.29 | -1.15 | 5.49 | 0.20 |
| Musculoskeletal | 4.27 | 1.86 | 0.20 | 2.30 | 0.59 | 7.94 | 0.02* |
| Low income | 4.93 | 2.12 | 0.21 | 2.33 | 0.74 | 9.13 | 0.02* |
| **Image** | | | | | | | |
| BILAG-2004 | 0.72 | 0.32 | 0.46 | 2.25 | 0.09 | 1.35 | 0.03* |
| M-SLEDAI | -0.85 | 0.78 | -0.23 | -1.09 | -2.39 | 0.69 | 0.28 |
| Age | -0.18 | 0.11 | -0.17 | -1.69 | -0.39 | 0.03 | 0.09 |
| Duration | -0.27 | 0.17 | -0.14 | -1.60 | -0.60 | 0.06 | 0.11 |
| Low income | 3.99 | 2.96 | 0.13 | 1.35 | -1.88 | 9.85 | 0.18 |
| Higher education | -2.54 | 2.38 | -0.10 | -1.07 | -7.25 | 2.17 | 0.29 |
| Muslim | -6.24 | 2.91 | -0.18 | -2.15 | -12.00 | -0.49 | 0.03* |
| Married | -1.08 | 2.42 | -0.04 | -0.45 | -5.88 | 3.71 | 0.66 |
| Renal | 1.01 | 2.36 | 0.04 | 0.43 | -3.66 | 5.67 | 0.67 |
| Hematological | 2.30 | 2.19 | 0.09 | 1.05 | -2.05 | 6.65 | 0.29 |
| **Treatment** | | | | | | | |
| BILAG-2004 | 0.15 | 0.12 | 0.28 | 1.26 | -0.08 | 0.38 | 0.21 |
| M-SLEDAI | -0.10 | 0.28 | -0.08 | -0.35 | -0.66 | 0.46 | 0.73 |
| Age | -0.06 | 0.04 | -0.16 | -1.66 | -0.13 | 0.01 | 0.10 |
| Duration | -0.02 | 0.06 | -0.03 | -0.31 | -0.14 | 0.10 | 0.76 |
| Higher education | -0.96 | 0.79 | -0.11 | -1.23 | -2.52 | 0.59 | 0.23 |
| **Mood** | | | | | | | |
| BILAG-2004 | 0.20 | 0.17 | 0.26 | 1.21 | -0.13 | 0.54 | 0.20 |
| M-SLEDAI | -0.10 | 0.41 | -0.05 | -0.25 | -0.91 | 0.70 | 0.80 |
| Age | -0.11 | 0.05 | -0.20 | -2.24 | -0.21 | -0.01 | 0.03* |
| Duration | -0.10 | 0.09 | -0.11 | -1.16 | -0.27 | 0.072 | 0.25 |
| Low socioeconomic | 2.07 | 1.35 | 0.13 | 1.53 | -0.61 | 4.75 | 0.13 |
| Haematological | 0.89 | 1.13 | 0.07 | 0.79 | -1.35 | 3.12 | 0.43 |

*significant p value at <0.05

validation of LupusQoL questionnaire has demonstrated good concurrent validity with SF-36, with very strong correlation coefficient ($r_s$ = 0.71 to 0.79) [11]. Similar results were obtained in the U.S. [35] and Spanish [36] validation studies.

There are inconsistencies in the relationship between QoL in SLE patients with disease activity [16]. The initial development and validation of SLEQoL in Singapore found no correlation with disease activity [12]. Other cross-cultural adaptation studies in Brazil also found poor correlation between SLEQoL and disease activity [37]. However, in our study, SLEQoL was found to be correlated with disease activity and the BILAG scores were the independent predictors of the physical function and image domains of SLEQoL. Cross-cultural validation study in Thailand also found significant correlation between disease activity measured with SLEDAI-2K with total SLEQoL-TH score, activity and symptom domains. Meanwhile, the Arabic SLEQoL demonstrates correlation between SLEDAI-2K with the total score, treatment and self-image domain of SLEQoL.

In contrast, poor correlation was found between LupusQoL and disease activity among our cohort of patients. However, the initial development [11]and other linguistic validation studies of the LupusQoL [38–40] have demonstrated a significant correlation with disease activity. This discrepancy can be explained by the difference in the recall period between these 2 instruments. It should also be noted that our study sample size was small and thus lack of heterogeneity as not many of our patients had active disease or considerable disease damage. Therefore, further larger studies are warranted to further include a wide spectrum of disease activity and damage of the SLE.

In our study, both SLEQoL and LupusQoL had no correlation with disease damage. This is consistent with the validation study in Brazil [37]. In contrast, SLEQoL cross-cultural validation study in Thailand [14] showed significant correlation between SLICC/ACR and the physical function domain while the Arabic version of SLEQoL showed significant correlation with physical, symptoms, treatment and self-image domains [15]. In contrast to our study, LupusQoL in other studies in the UK [11], US [35], China [39], Mexico [41], Iran [42] and Spanish [43] the were found to be able to discriminate based on damage (SDI scores damage). However, majority of these studies demonstrates weak correlations with the damage index.

Our study found that MSK involvement was the independent predictor of physical function, activity and symptom domains in M-SLEQoL. Meanwhile, MSK and NPSLE were the independent predictor of low QoL in fatigue domain of M-LupusQoL. Both MSK and NPSLE were indeed associated with lower QoL in various other studies [44–46].

## Strength and limitations

Our study was the first to perform the validation and cross-cultural adaption of the two commonly used SLE-specific health related questionnaires in Malay language. We have shown that both M-SLEQoL and M-LupusQoL had good internal consistencies, acceptable factor analysis and concurrent validity with the M-SF36. However, there are several limitations in our study. Our study was a cross-sectional study so we did not perform test-retest reliability. During the cognitive debriefing, we only obtained subjective feedback from the respondents and did not measure the Item-face validity index (I-FVI) and (Scale-face validity index) S-FVI. The sample size in our study also was small so we did not perform confirmatory factor analysis (CFA). Instead we use the exploratory factor analysis (EFA) for preliminary determination of the questionnaires' construct validity [47]. Therefore, a larger study is needed in the future to further confirm the validity in a new data set of patients. Our study also could not establish any responsiveness of the M-SLEQoL and M-LupusQoL to changes. In addition, many other possible confounding variables that might influence QoL of the patients were not measured in our

study. For instance co-morbidity fibromyalgia has been shown to be an independent and important HRQoL predictor [16, 41] were not measured in this study. Different types of treatment and their side-effects were also not studied.

## Conclusion

The results of this study provide evidence regarding the reliability and validity of the Malays version of SLEQoL and LupusQoL questionnaires in our SLE population. However, M-Lupus-QoL has better discriminant validity as it functions independently from measures of disease activity and damage. The QoL of patients with musculoskeletal (MSK) involvement were mostly affected as it was associated with lower M-SLEQoL in physical function, activity and symptom domains, and fatigue domain in M-LupusQoL.

## Supporting information

**S1 File. Item-content validity index (I-CVI) and scale content validity index of M-SLEQoL and M-LupusQoL.**
(XLSX)

**S2 File. Raw data.**
(XLSX)

**S1 Table. Cronbach's alpha for the total domain and its sub-items of SLEQoL.**
(DOCX)

**S2 Table. Cronbach's alpha for the total scale and its subscales' and item analysis of Lupus-QoL.**
(DOCX)

**S3 Table. Eigen values for M-SLEQoL.**
(DOCX)

**S4 Table. Eigen values for M-LupusQoL.**
(DOCX)

**S5 Table. Exploratory Factor Analysis of 40 items in M-SLEQoL (values below 0.4 are suppressed).**
(DOCX)

**S6 Table. Exploratory factor analysis of the 34 items in M-LupusQoL (values below 0.4 are suppressed).**
(DOCX)

**S1 Appendix. SLEQoL Malay version.**
(DOC)

**S2 Appendix. LupusQoL Malay version.**
(DOCX)

## Author Contributions

**Conceptualization:** Nur Aqeelah Ahmad Pouzi, Syahrul Sazliyana Shaharir, Azmi Mohd Tamil, Ruslinda Mustafar, Suhaida Ahmad Maulana, Eashwary Mageswaren.

**Data curation:** Nur Aqeelah Ahmad Pouzi, Syahrul Sazliyana Shaharir.

**Formal analysis:** Nur Aqeelah Ahmad Pouzi, Azmi Mohd Tamil.

**Methodology:** Nur Aqeelah Ahmad Pouzi, Syahrul Sazliyana Shaharir, Azmi Mohd Tamil, Ruslinda Mustafar.

**Project administration:** Nur Aqeelah Ahmad Pouzi, Syahrul Sazliyana Shaharir, Ruslinda Mustafar, Suhaida Ahmad Maulana, Eashwary Mageswaren.

**Supervision:** Syahrul Sazliyana Shaharir, Azmi Mohd Tamil, Ruslinda Mustafar, Suhaida Ahmad Maulana, Eashwary Mageswaren, Wan Syamimee Wan Ghazali.

**Validation:** Azmi Mohd Tamil, Wan Syamimee Wan Ghazali.

**Writing – original draft:** Nur Aqeelah Ahmad Pouzi, Azmi Mohd Tamil.

**Writing – review & editing:** Syahrul Sazliyana Shaharir, Ruslinda Mustafar, Suhaida Ahmad Maulana, Eashwary Mageswaren, Wan Syamimee Wan Ghazali.

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
