## [Decision Letter · Decision Letter 0]

31 Jan 2023

PONE-D-22-32067Validation and the associated factors of the systemic lupus erythematosus-specific health-related quality of life questionnaires (SLEQoL and LupusQoL) in MalaysiaPLOS ONE

Dear Dr. Shaharir,

Thank you for submitting your manuscript to PLOS ONE. After careful consideration, we feel that it has merit but does not fully meet PLOS ONE’s publication criteria as it currently stands. Therefore, we invite you to submit a revised version of the manuscript that addresses the points raised during the review process.

Please kindly address all comments by the reviewers.

We look forward to receiving your revised manuscript.

Kind regards,

Shazlin Shaharudin

Academic Editor

PLOS ONE

Journal Requirements:

4 Please review your reference list to ensure that it is complete and correct. If you have cited papers that have been retracted, please include the rationale for doing so in the manuscript text, or remove these references and replace them with relevant current references. Any changes to the reference list should be mentioned in the rebuttal letter that accompanies your revised manuscript. If you need to cite a retracted article, indicate the article’s retracted status in the References list and also include a citation and full reference for the retraction notice.

Reviewers' comments:

Reviewer's Responses to Questions

**Comments to the Author**

1. Is the manuscript technically sound, and do the data support the conclusions?

Reviewer #1: No

Reviewer #2: Yes

Reviewer #3: Yes

2. Has the statistical analysis been performed appropriately and rigorously? 

Reviewer #1: No

Reviewer #2: Yes

Reviewer #3: Yes

3. Have the authors made all data underlying the findings in their manuscript fully available?

Reviewer #1: Yes

Reviewer #2: Yes

Reviewer #3: Yes

4. Is the manuscript presented in an intelligible fashion and written in standard English?

Reviewer #1: No

Reviewer #2: No

Reviewer #3: Yes

5. Review Comments to the Author

Reviewer #1: The writing is unclear, has repetition, and is not consistent. It needs to be clarified which questionnaire the author wanted to validate since, in the title, the full name of the questionnaire refers to SLEQol. Still, two questionnaires were mentioned by the author: SLEQol and LupusQol. The author should focus on the validated questionnaire, and the whole paper should only be about the validation study. It is recommended to separate the validation study and relationship study.

In the validation study, the title of the questionnaire is significant. However, the questionnaire’s title in this manuscript is inconsistent (SLEQoL, LupusQoL, M-SLEQoL, and M-LupusQoL). The methodology was also unclear. The author did not mention when they used the SLEQol to gather information for the validation study. The author did not state the sampling method and the sample size calculation.

Validity: For content and face validity, the author should provide the results of I-CVI (Item-content validity index), S-CVI (Scale-Content validity Index), I-FVI (Item-face validity index), and S-FVI (Scale-face validity index). The author should report the Eigenvalues and the number of factors and items produced by the EFA. The author did not provide the analysis for confirmatory factor analysis (CFA), which is the primary analysis for construct validity. Thus, the validation process of the questionnaire still needs to be fulfilled, and the questionnaire still needs to be validated. Since the questionnaire has not been validated, studies on associated factors using invalid SLEQoL are questionable.

Reviewer #2: Thank you for the opportunity to review your manuscript. The review may seem strict; however, if you can address the comment. I feel that it would significantly improve the quality and readability of your manuscript and place it in bedstead for publication and citing. Please note that there are no page or line numbers, and I find it difficult to mention my comments.

Abstract:

1. objectives

• “To assess the reliability and validity of SLE QoL questionnaires, SLEQoL and LupusQoL in Malay language, and to identify the factors associated with QoL among SLE patients in Malaysia.” The sentence is not complete and clear, please rewrite it. (Page 2)

• please spell words out in full and provide the abbreviation in parentheses. (Page 2)

2. Conclusion

• “Musculoskeletal and NPSLE were the independent predictors of poor QoL.” Please clarify it. (Page 2)

Introduction:

• put the clinical phenotype instead of clinical phenotype. (Page 3)

• put the United State instead of United State. (Page 3)

• “The most-widely used SLE disease-specific PRO tools is LupusQoL which were

developed in the UK” rewrite and correct it grammatically. (Page 3)

• Please correct the spell of randomised–controlled. (Page 4)

• “validated English language among Singaporean Chinese patients.” Put in before English language. (Page 4)

Materials and methods

Patients:

• Please add information about the number of your sample and patients. (Page 5)

Results

• Employment status in table 2, please write the other status information

• In table 3, please specify interquartile ranges

• Please complete table 4 with Descriptive statistics

Discussion

• Please add suitable articles to the words (such as a poor correlation, a cross-cultural)

Reviewer #3: The manuscript was well written.

Introduction is well written. Authors explain justified the problem statement well.

Methodology was explained in detail to allow replication. Proper flow of validation techniques was used.

Results were described phase by phase based on the analyses conducted.

Discussion was well written. Authors compared their findings with other studies beside explaining the discrepancies.

Conclusion was too short.

Based typing and grammar, there is a very minimum grammatical error and no need for proofreading.

There are also typing error :

page 9, last line, andhas for and has,

page 19: second line under the subtopic of Concurrent validity (SD-36 instead of SF-36).

Other than that, I would suggest the authors to use standardized method of writing money currency, MYR instead of RM.

6. PLOS authors have the option to publish the peer review history of their article (what does this mean?). If published, this will include your full peer review and any attached files.

Reviewer #1: No

Reviewer #2: No

Reviewer #3: No

---

## [Author Response · Author response to Decision Letter 0]

16 Apr 2023

Thank you for the reviewers’ comments. We have revised the manuscript according to the reviews and below are the responses to the comments and suggestions.

Reviewer 1

1. The writing is unclear, has repetition, and is not consistent. It needs to be clarified which questionnaire the author wanted to validate since, in the title, the full name of the questionnaire refers to SLEQol. Still, two questionnaires were mentioned by the author: SLEQol and LupusQol. 

We have clarified in the objective of this study as stated in the abstract and the main manuscript (page 4):

To assess the reliability and validity of two disease-specific questionnaires that assess the quality of life (QoL) among patients with Systemic Lupus Erythematosus (SLE); SLEQoL and LupusQoL in Malay language.

2. The author should focus on the validated questionnaire, and the whole paper should only be about the validation study. It is recommended to separate the validation study and relationship study.

We have separated the validation part and the relationship part in the methodology and result in this manuscript. However, we did not separate them into two manuscripts to allow a continuity and better understanding on the impact of the disease to the quality of life in our patients.

3. In the validation study, the title of the questionnaire is significant. However, the questionnaire’s title in this manuscript is inconsistent (SLEQoL, LupusQoL, M-SLEQoL, and M-LupusQoL). 

Thank you for the review and comments. We have revised the manuscript and have consistently state the original version of the questionnaires as SLEQoL and Lupus QoL, while the Malay-translated version are M-SLEQoL and M-Lupus QoL (as highlighted in the manuscript)

4. The methodology was also unclear. The author did not mention when they used the SLEQol to gather information for the validation study. 

Both of the questionnaires were distributed to patients using hardcopies or google forms in outpatient clinic from June 2021 until April 2022. We have also clarified the methodology in the highlighted text of the Material & Methods section.

5. The author did not state the sampling method and the sample size calculation.

The sampling method was convenience sampling and this has been added in the Material & Methods (page 5). 

Sample size calculation has been added in the Material & Methods section (page 5).

6. Validity: For content and face validity, the author should provide the results of I-CVI (Item-content validity index), S-CVI (Scale-Content validity Index), I-FVI (Item-face validity index), and S-FVI (Scale-face validity index). 

 Content validity was performed by the expert panels of the researchers which consists of 2 rheumatologists, and an academician public health specialist experienced in the validation of instruments. They have reviewed the content and decided that all the items were relevant and important. The Item-content validity index (I-CVI) and scale content validity index of both questionnaires were 1 (Supplementary File 1). The expert panels have also reviewed the translated version of the questionnaires and found that they were similar as the original version. 

Face validity was done through cognitive debriefing of the questionnaires in 10 healthy subjects and 10 SLE patients to determine items clarity and comprehensibility. They were asked for their views regarding the phrasing and content of the instruments. All subjects understood the translated version and did not encounter problems in answering the questionnaire. However, we did not record the subjects’ rates on the items in the questionnaires and hence the Item-face validity index (I-FVI) and (Scale-face validity index) S-FVI were not calculated.

We have added this in the Material & Methods section: Content, translation and face validity of the SLEQoL and LupusQoL in Malay language (highlighted in page 7). We have also stated this limitation in the “Strength and Limitation” section section.

7. The author should report the Eigenvalues and the number of factors and items produced by the EFA. 

The results have been added in the manuscript, as highlighted in the text. We have also added Table S3 which illustrates the Eigen values for M-SLEQoL and Table S4 shows Eigen values for M-LupusQoL

8. The author did not provide the analysis for confirmatory factor analysis (CFA), which is the primary analysis for construct validity. Thus, the validation process of the questionnaire still needs to be fulfilled, and the questionnaire still needs to be validated. Since the questionnaire has not been validated, studies on associated factors using invalid SLEQoL are questionable.

Since the sample size required to perform CFA is 200, we did not perform the analysis. The conventional method ie exploratory factor analysis (EFA) was performed instead to test for construct validity, which is also an acceptable method to preliminary assess construct validity (Mohamad Adam B., Hon YK, & Lee KY. (2022). A Step By Step Guide To Questionnaire Validation Research. Kuala Lumpur, Malaysia: Institute for Clinical Research, NIH MY. ISBN: 9789671694060 (MY) DOI: 10.5281/zenodo.6801209) 

We did acknowledge the limitation of our study which used EFA in construct validity analysis in the Strength and Limitation section (page 22) and recommended that a future larger study should be performed to further confirm the construct validity using the confirmatory factor analysis (CFA).

Reviewer 2: 

Thank you for the opportunity to review your manuscript. The review may seem strict; however, if you can address the comment. I feel that it would significantly improve the quality and readability of your manuscript and place it in bedstead for publication and citing. Please note that there are no page or line numbers, and I find it difficult to mention my comments.

Abstract:

1. objectives

• “To assess the reliability and validity of SLE QoL questionnaires, SLEQoL and LupusQoL in Malay language, and to identify the factors associated with QoL among SLE patients in Malaysia.” The sentence is not complete and clear, please rewrite it. (Page 2)

• please spell words out in full and provide the abbreviation in parentheses. (Page 2)

Objectives were revised to: To assess the reliability and validity of two disease-specific questionnaires that assess the quality of life (QoL) among patients with Systemic Lupus Erythematosus (SLE); SLEQoL and LupusQoL in Malay language. This study will also identify the factors associated with poor QoL among SLE patients in Malaysia.

2. Conclusion

• “Musculoskeletal and NPSLE were the independent predictors of poor QoL.” Please clarify it. (Page 2)

Based on the multivariable regression analysis, MSK involvement was the independent factor of lower QoL in multiple domains including physical function, activity, symptom domains in M-SLEQoL and fatigue in M-LupusQoL. Therefore, the conclusion has been revised to:

“SLE patients with MSK involvement are at risk of poor QoL in multiple domains including physical function, activity, symptoms and fatigue”.

3. Introduction:

• put the clinical phenotype instead of clinical phenotype. (Page 3)

• put the United State instead of United State. (Page 3)

• “The most-widely used SLE disease-specific PRO tools is LupusQoL which were

developed in the UK” rewrite and correct it grammatically. (Page 3)

• Please correct the spell of randomised–controlled. (Page 4)

• “validated English language among Singaporean Chinese patients.” Put in before English language. (Page 4)

Thank you for the reviews. Corrections have been made.

4. Materials and methods

Patients:

• Please add information about the number of your sample and patients. (Page 5)

Sample size calculation has been added in page 5

5. Results

• Employment status in table 2, please write the other status information

• In table 3, please specify interquartile ranges

• Please complete table 4 with Descriptive statistics

Corrections have been made (highlighted in Table 2, Table 3 and Table 4). 

6. Discussion

• Please add suitable articles to the words (such as a poor correlation, a cross-cultural)

Discussion has been revised as per suggestion by the reviewer

Reviewer 3

The manuscript was well written.

Introduction is well written. Authors explain justified the problem statement well.

Methodology was explained in detail to allow replication. Proper flow of validation techniques was used.

Results were described phase by phase based on the analyses conducted.

Discussion was well written. Authors compared their findings with other studies beside explaining the discrepancies.

1.Conclusion was too short

Conclusion has been revised to : 

The results of this study provide evidence regarding the reliability and validity of the Malays version of SLEQoL and LupusQoL questionnaires in our SLE population. However, M-LupusQoL has better discriminant validity as it functions independently from measures of disease activity and damage. The QoL of patients with musculoskeletal (MSK) involvement were mostly affected as it was associated with lower M-SLEQoL in physical function, activity and symptom domains, and fatigue domain in M-LupusQoL.

2. Based typing and grammar, there is a very minimum grammatical error and no need for proofreading.

There are also typing error : page 9, last line, and has for and has,

page 19: second line under the subtopic of Concurrent validity (SD-36 instead of SF-36).

Thank you for the review. Corrections have been made.

3. Other than that, I would suggest the authors to use standardized method of writing money currency, MYR instead of RM.

Thank you for the review. Corrections have been made.

---

## [Editor Report · Decision Letter 1]

25 Apr 2023

Validation and the associated factors of the Malay version of systemic lupus erythematosus-specific health-related quality of life questionnaires (SLEQoL and LupusQoL)

PONE-D-22-32067R1

Dear Dr. Shaharir,

We’re pleased to inform you that your manuscript has been judged scientifically suitable for publication and will be formally accepted for publication once it meets all outstanding technical requirements.

Kind regards,

Shazlin Shaharudin

Academic Editor

PLOS ONE
---

## [Editor Report · Acceptance letter]

5 May 2023

PONE-D-22-32067R1 

Validation and the associated factors of the Malay version of systemic lupus erythematosus-specific health-related quality of life questionnaires (SLEQoL and LupusQoL). 

Dear Dr. Shaharir:

I'm pleased to inform you that your manuscript has been deemed suitable for publication in PLOS ONE. Congratulations! Your manuscript is now with our production department. 

Kind regards, 

on behalf of

Dr. Shazlin Shaharudin 

Academic Editor

PLOS ONE